

# Representativeness of single lidar stations for zonally averaged ozone profiles, their trends and attribution to proxies

Christos Zerefos[1,2], John Kapsomenakis[1], Kostas Eleftheratos[3], Kleareti Tourpali[4], Irina Petropavlovskikh[5], Daan Hubert[6], Sophie Godin-Beekmann[7], Wolfgang Steinbrecht[8], Stacey Frith[9], Viktoria Sofieva[10], Birgit Hassler[11]

[1]Research Centre for Atmospheric Physics and Climatology, Academy of Athens, Athens, Greece
[2]Navarino Environmental Observatory (N.E.O), Messinia, Greece
[3]Department of Geology and Geoenvironment, National and Kapodistrian University of Athens, Greece
[4]Department of Physics, Aristotle University of Thessaloniki, Greece
[5]Cooperative Institute for Research in Environmental Sciences, University of Colorado, Boulder, CO, USA
[6]Royal Belgian Institute for Space Aeronomy (BIRA-IASB), Brussels, Belgium
[7]Laboratoire Atmosphère Milieux Observations Spatiales, Centre National de la Recherche Scientifique,
Université de Versailles Saint-Quentin-en-Yvelines, Université Pierre et Marie Curie, Guyancourt, France
[8]Deutscher Wetterdienst, Hohenpeißenberg, Germany
[9]NASA Goddard Space Flight Center, Silver Spring, MD, USA
[10]Finnish Meteorological Institute, Helsinki, Finland
[11]Deutsches Zentrum für Luft- und Raumfahrt, Institut für Physik der Atmosphäre, Oberpfaffenhofen, Germany

*Correspondence to*: Christos S. Zerefos (zerefos@geol.uoa.gr)

**Abstract.** The paper is focusing on the representativeness of single lidar stations for zonally averaged ozone profile variations over the middle and upper stratosphere. From the lower to the upper stratosphere, ozone profiles from single or grouped lidar stations correlate well with zonal means calculated from (Solar Backscatter Ultraviolet Radiometer (SBUV) overpasses. The best representativeness is found within a few degrees of latitude north or south of any lidar station. The latitude range with significant correlation coefficients (>0.4) spans about ±10° in the mid-stratosphere (around 30 hPa) and becomes much larger in the upper stratosphere (around 2 hPa), where it spans a large part of the entire globe. The paper includes also a multiple linear regression analysis on the relative importance of proxy time series for explaining variations in the vertical ozone profiles. Studied proxies represent variability due to influences outside of the earth system (solar cycle), variability due to dynamic processes (the Quasi Biennial Oscillation (QBO), the Arctic Oscillation (AO), Antarctic Oscillation (AAO), El Niño Southern Oscillation (ENSO)), due to volcanic aerosol (Aerosol Optical Depth (AOD)), due to tropopause height changes (including global warming) and due to manmade contributions to chemistry (Equivalent Effective Stratospheric Chlorine (EESC)). Ozone trends are estimated, with and without removal of proxies, from the total available 1980 to 2015 SBUV record. Except for the chemistry related proxy (EESC), the use of the other proxies does not alter the significance of the estimated long-term trends. At heights above 10 hPa an "inflection point" between 1997 and 1999 marks the end of significant negative ozone trends, followed by a recent period of positive ozone change over the period 1998-2015. At heights below 15 hPa the pre-1998 negative ozone trends tend to become insignificant as we move towards 2015.





## 1 Introduction

Two recently published papers (Steinbrecht et al., 2017; Weber et al., 2017) show that total ozone and ozone profile trends are consistent with earlier WMO (2014) findings. Despite the addition of 4 more years since WMO (2014), Weber et al. (2017) show that for most datasets and regions the trends in total ozone, since stratospheric halogens reached their maximum around 1997, are not significantly different from zero. In the case of ozone profile trends, however, Steinbrecht et al. (2017) confirmed increasing trends in the upper stratosphere (2 hPa) as first reported in WMO (2014). Moreover, due to improved data sets and longer records, the uncertainty in the

profile trends reported by Steinbrecht et al. (2017) was reduced by a factor of 2 compared to Harris et al. (2015).

In this work we have analysed SBUV (McPeters et al., 2013; Frith et al., 2017) and lidar ozone profile data from the Network for the Detection of Atmospheric Composition Change (NDACC) as part of the Long-term Ozone Trends and Uncertainties in the Stratosphere (LOTUS) project (http://igaco-o3.fmi.fi/LOTUS/index.html). The

project aims at providing support and input to the WMO/UNEP 2018 Ozone Assessment for a better understanding of ozone trends and their significance as a function of altitude and latitude, nearly 20 years after the peak of ozone depleting substances in the stratosphere. Among the objectives of the LOTUS initiative is the improvement of our understanding of all sources of uncertainties in estimated trends and regression methods. In this work we provide a new look at the uncertainties involved in the representativeness of single (lidar) stations

for zonally averaged layer ozone. We then look at ozone trends and at the hierarchy of proxies commonly used in statistical ozone trend analyses. We try to provide a better understanding of uncertainties and to quantify the effect of stratospheric climatology and chemistry on the estimated profile trends.

## 2 Data, analysis and methods

### 2.1 Satellite data

Solar Backscatter UltraViolet (SBUV) version 8.6 station overpass satellite data for the period 1980-2015 have been analysed through this work. The SBUV observing system consists of a series of instruments that measure ozone profiles from the ground to the top of the atmosphere (e.g. DeLand et al., 2012; McPeters et al., 2013). Measurements are provided as partial column ozone amounts in Dobson Units (DU). We have analysed ozone data for 7 pressure layers as shown in Table 1.


The satellite data come from all SBUV type instruments with data availability from 11/1978 to the present (see Table 2 for details). Three versions of the SBUV instrument are used in the series, but the fundamental measurement technique is the same over the evolution of the instrument from BUV to SBUV/2 (Bhartia et al., 2013). Satellite overpasses over a number of ground-stations are available for each day from the website

ftp://toms.gsfc.nasa.gov/pub/sbuv/AGGREGATED/. Daily averages have been calculated by averaging the measurements from all available satellite instruments. Then monthly means were derived following the instructions provided at https://acd-ext.gsfc.nasa.gov/Data_services/merged/instruments.html. Additional SBUV data used in the present work include 5 degree of latitude zonal means taken from ftp://toms.gsfc.nasa.gov/pub/MergedOzoneData/Ind_Inst_HDF/ (McPeters et al., 2013).



**2.2 Lidar data**

Monthly mean ozone profiles from ground-based lidar instruments were obtained by averaging daily profiles from the NDACC Database at ftp://ftp.cpc.ncep.noaa.gov/ndacc/station/. Data for lidar stations with long term measurements, namely, Hohenpeißenberg (47.8° N, 11.0° E), Haute Provence (43.9° N, 5.7° E) and Table Mountain (34.4° N, 117.7° W) in the northern mid-latitudes, Mauna Loa (19.5° N, 155.6° W) in the tropics and

Lauder (45.0° S, 169.7° E) in the southern mid-latitudes were taken from the NDACC NASA-Ames format files. It should be noted here that all lidar measurements are given as number density (molec.cm-3) versus altitude. From these measurements the column densities in m-atm-cm (D.U.) were calculated for the corresponding SBUV layers using the equation:

$$\text{Column density (in D.U.)} = \sum_{z0}^{z1}[O_3 \text{ (in molecules/cm}^3)] * \frac{\Delta z \text{ (cm)}}{2.69 \times 10^{16}} \tag{1}$$

Where $z0$ is the base and $z1$ is the top of each SBUV layer and $\Delta z$ is the height interval between two successive lidar measurements. The relation between height and atmospheric pressure is derived from ERA-interim reanalysis data interpolated to each station.

**3 Representativeness of single station ozone profiles in comparison to zonal means**

The comparison between lidar and SBUV station overpasses on common days throughout the record was first done on the basis of deseasonalized monthly mean lidar and SBUV ozone profiles. Figure 1a shows the resulting correlation coefficients. All correlations are statistically significant at the 99.99% confidence level. Recalculation after removing the strong trends before the 1998 does not alter the significance. The average distance between

the subsatellite points and lidar station were 500 km at middle latitude stations (Hohenpeißenberg, Haute Provence and Lauder) and 700 km for lower latitude lidar stations (Table Mountain and Mauna Loa) with collocation time criterion being sub daily.

The correlation coefficients in Figure 1a show a structure in the vertical. This is a result of using different

instruments and different sampling times (SBUV data are daytime drifting orbit, lidar data are night-time). The declining signal to noise ratio for the lidars above 35 to 40 km also plays a role. Larger atmospheric variability at higher latitudes tends to increase correlations, e.g. at Lauder and Hohenpeißenberg, as does the very regular and large QBO signal at Mauna Loa.

To check the effect of different sampling, we also calculated the correlation between monthly mean SBUV overpasses averaged over all ≈30 days in a month, with SBUV overpasses averaged only over those days when lidar measurements were available. These results are presented in Figure 1b. Now the vertical structure is reduced, indicating that the drop above 35 km in Figure 1a is due to instrumental differences between SBUV and the lidars. The drop in correlation around 32 km in Figure 1b indicates atmospheric variability that is sampled

differently, when measurements are available only on the lidar dates. Interestingly, this variability seems to occur predominantly at the mid-latitude stations, not at Mauna Loa.



We now come to the question of the representativeness of ozone monthly means at single stations compared to 5° latitude zonal means calculated for SBUV. Figure 2a shows the profiles of correlations between SBUV

monthly 5° zonal means and SBUV overpass monthly means at the lidar locations. Again, all correlation coefficients are large (0.70 to 0.95) and highly significant (99.99%). The increase of the correlations with altitude is in part due to the larger trends at higher latitudes, which increase the signal to noise ratio in the time series, and increase correlations, especially on long time scales.

Finally, Figure 2b gives the correlation between SBUV monthly zonal means and lidar station monthly means. These correlation coefficients are substantially reduced, but are still statistically significant, except at Table Mt above 10hPa.

The previous figures help to explain the observed range of correlations in Figure 2b: As shown in Figure 2a for

SBUV data, the correlation between station monthly means and zonal means drops by 0.1 to 0.2 from a perfect value of 1. This is largely due to longitudinal variations, which are smallest at lower latitudes / Mauna Loa. Figure 1b, again on the basis of SBUV data, then indicates that the sparse temporal sampling of the lidars, also leads to drops in the correlation by 0.1 to 0.3, compared to the perfect correlation value of 1. Again, this is less critical at Mauna Loa, where either better sampling or lower temporal variability (or both) gives the highest

correlations.

Figure 1a indicates that instrumental differences between the lidars and SBUV (different vertical resolution, different accuracy, different long-term stability) result in correlations between 0.4 and 0.8 for monthly mean data with comparable sampling. Reduced temporal sampling by the lidars (compare Figure 1b), and longitudinal

variations not sampled by a single station (compare Figure 2a), together explain the reduced correlations, 0.2 to 0.6, between lidar monthly means and SBUV zonal means in Figure 2b.

A further look at the spatial distribution of correlation coefficients between single SBUV overpasses at lidar stations (or station groups) and SBUV overpasses for all 633 stations available at the website

ftp://toms.gsfc.nasa.gov/pub/sbuv/AGGREGATED/ is given in Figures 3 to 6. The fairly good "zonal representativeness" of the stations is obvious from the color scale. Also, as we move higher in altitude, high correlations are found even at distances exceeding 1000 km, and spanning almost the entire globe. In layer 14 each station is representative for most of the globe. One reason for this global representativeness at the highest layer is the increased and global importance of long-term ozone trends, and possibly other common sources of

ozone variation, at the uppermost levels. This is investigated in more detail in the following section.

## 4 The role of proxies in the variability of ozone

A number of proxies have been used to explain the variability in space and time of the vertical ozone distribution, superimposed to the dominating annual cycle (Zerefos et al., 1992; Reinsel et al., 2002; Newchurch et al., 2003; Reinsel et al., 2005; Zanis et al., 2006; Nair et al., 2013; Frith et al., 2014; Harris et al., 2015;

Steinbrecht et al., 2017; Weber et al., 2017; WMO 2007, 2011, 2014). Each proxy reflects ozone variability in a different way. For instance, the El Niño Southern Oscillation (ENSO) has specific geographic patterns of



influence in total ozone and its effect is confined in the upper troposphere/lower stratosphere (Zerefos et al., 1992). The Quasi Biennial Oscillation (QBO) is influencing ozone from the middle stratosphere down to the troposphere with a phase progressing both in height and latitude at rates of about 1 km per month vertically, and by about 4 degrees of latitude per month (Zerefos, 1983).

The proxies can be grouped into the following categories: (1) Dynamical proxies. These include: the Quasi Biennial Oscillation (QBO), the El Niño Southern Oscillation (ENSO), the Arctic Oscillation (AO), the Antarctic Oscillation (AAO) and Tropopause Pressure. (2) Extraterrestrial proxies. This is primarily the 11-year solar cycle and (3) stratospheric composition proxies, typically stratospheric aerosol optical depth (e.g. at 550 nm) and equivalent effective stratospheric chlorine (EESC).

In order to investigate both qualitatively and quantitatively the attribution of ozone variations to the different proxies, we have used a multi-linear regression method, as described in the following paragraph.

### 4.1 Regression analysis model

Multivariate linear regression (MLR) analysis has been applied both to SBUV and lidar data sets (e.g. WMO, 2011; Nair et al., 2013; Harris et al. 2015). Historically, long-term trends in ozone have been investigated with the use of simple linear trends. More sophisticated methods allowing for the estimation of a change in the long term trend (such as the PWLT), or using directly the EESC (Equivalent Effective Stratospheric Chlorine) as a proxy to estimate the rate of change in ozone losses due to the evolution of ODSs, have been used e.g. by Reinsel et al. (2005) or in the ozone assessments (WMO, 2014).

In this work we have used the statistical model in two ways, using either (a) the Piecewise Linear Trend (PWLT) method, with January 1998 selected as inflection point, or (b) EESC as a proxy. The MLR regression model, in each case, was applied at all seven pressure levels and for the different zonal belts / stations. Our MLR model takes the general form:

$$\Delta O_3(t) = \mu + a_{trend}Trend + a_{qbo}QBO(t) + \alpha_{solar}SOLAR(t) + \alpha_{enso}ENSO(t) + \alpha_{AO}AOI(t) +$$
$$a_{trop_{pres}}trop_{pres}(t) + a_{volcanic}AOD(t) + N(t) \tag{2}$$

Where the term $a_{trend}Trend$ corresponds to either (a) a PWLT or (b) the EESC proxy:

(a) $\alpha_{tr1}T_1(t) + \alpha_{tr2}T_2(t)_{(t=0\ for\ t<1998)}$, in the case of the PWLT runs, with $T_1$ and $T_2$ accounting for pre- and post- 1998 linear trends, and $T_2$ set to zero before 1/1998,

(b) $\alpha_{eesc}EESC(t)$, for the runs with EESC as a proxy, corresponding to 3-years age of air (Newman et al., 2007).

Overall, $\Delta O_3(t)$ is the time-series of ozone anomalies in percent (%) for a particular month $t$. Data are deseasonalised prior to the analysis, by removing the long-term monthly average (1980-2015) for each calendar month (January, February, … December).





The other terms are:

- *μ* corresponds to a constant term,
- For the *QBO* term, equatorial zonal winds at 30 and 50 hPa as given by the standardized NOAA –CPS indices for 30 and 50 hPa, were used (http://www.cpc.ncep.noaa.gov/data/indices/).
- *SOLAR* accounts for the solar cycle effect in ozone, using the 10.7 cm wavelength solar radio flux (F10.7) as a proxy.
- Similarly, *ENSO* accounts for the ENSO effect on ozone, using the MEI (Multivariate ENSO Index) as a proxy http://www.esrl.noaa.gov/psd/enso/mei/table.html.
- The *AOI* term is used to describe the Arctic (or Antarctic) Oscillation effect on ozone. The AO Index is used for the Northern hemisphere and the AAO Index for the Southern hemisphere. Both come from NOAA: http://www.cpc.ncep.noaa.gov/products/precip/CWlink/daily_ao_index/teleconnections.shtml
- *trop_{pres}* is the term used to describe the effect of tropopause changes on ozone. This index is constructed from NCEP re-analysis tropopause pressures. It is filtered to remove ENSO, solar, QBO, long-term trend and volcanic effects. The index is calculated separately for every data set used here, either as a zonal mean for the SBUV zonal averages, or for each station (lidar or SBUV overpasses).
- *AOD* is used to describe volcanic effects: The zonal mean 550 nm Stratospheric Aerosol Optical Depth is used from http://data.giss.nasa.gov/modelforce/strataer/. As the index stops at the end of 2012, this year was repeated until the end of ozone record.
- *N(t)* is the residual noise series, assumed to be an autoregressive AR(1) time series with $N(t) = \varphi N(t\text{-}1) + \varepsilon(t)$, where *ε(t)* is an uncorrelated series, with weights inversely proportional to the monthly residual variances, in which the uncertainties of the monthly averages were taken into account.

Trends and errors (especially for the PLWT runs) are calculated as in Reinsel et al. (2002) and the results are given in % of the respective long-term mean.

**4.2 MLR results and discussion**

Figure 7 shows the amplitude [maximum value – minimum value / 2] of ozone variability attributed to each proxy for the 7 vertical layers and for Hohenpeißenberg, Mauna Loa and Lauder. Amplitudes are given in % of the long-term ozone mean.

The upper panel of Figure 7 shows results for Hohenpeißenberg as a northern mid-latitude example, the middle panel for Mauna Loa as a tropical latitude and the bottom panel Lauder as a southern mid-latitude example. The left plots refer to monthly mean SBUV overpasses for the whole period 1980-2015, the plots in the middle refer to SBUV data for the period common with Lidar measurements, and the right plots refer to the Lidar monthly mean ozone profiles. The amplitude of QBO related variations below 10 hPa, down to 40 hPa, is on the order of 2% of the mean. The smallest QBO amplitudes are found in the uppermost layers 13 and 14 (0.5% of the mean or less). We should point out, that according to Kramarova et al. (2013) the coarse vertical resolution of SBUV (and the decreasing altitude resolution of the lidars above 35 to 40 km) can induce errors in the amplitude of QBO related ozone anomalies on the order of 1% at heights between 10 and 1 hPa. However, for trend analysis this is not expected to introduce any significant effect.






The footprint of the solar cycle is clearly seen in the middle and upper stratosphere with amplitudes around 2% of the mean. The amplitude of AO (AAO in the Southern hemisphere) in the zonal mean is about 1% of the mean. At individual levels or stations it can be as high as 4% of the mean. The contribution of ENSO (MEI) is typically less than 1% of the mean at Hohenpeißenberg and Lauder, but up to 4% for the Mauna Loa SBUV data.

The effect of tropopause height variations is most evident in the lower stratospheric layer 8, where it reaches 4% for the SBUV data at Lauder and Hohenpeißenberg, but only 2% for the lidar data. The lidars have better altitude resolution than SBUV in the mid and lower stratosphere, and do not include a substantial contribution from levels below 40 hPa / 26 km. In the upper levels, tropopause height related ozone variations generally decrease.

Transitional effects from large AOD of volcanic origin (El Chichon, Pinatubo) can contribute substantial ozone variability, 4 to 6% of the mean, for shorter time periods (2 to 3 years after the volcano).

Finally, the EESC proxy representing halogen chemistry carries the largest and most significant ozone variations, up to 5% of the mean in the upper stratosphere. These results are in general agreement with previous

results by Nair et al. (2013) and Kirgis et al. (2013).

Important aspects of Figure 7 are: (a) the overall ozone variations explained all proxies together, typically 5 to 12% of the mean, are very similar for the 1980 to 2015 SBUV overpass time series, for the shorter SBUV – lidar common period, and for the lidar station time series. (b)For many individual proxies the attributed ozone

variability is also very similar for all three station related time series. Together with the correlations shown in the previous section, this means that ground-based instruments at single stations can provide representative information about ozone trends (EESC) and ozone variations related to the QBO, the solar cycle, ENSO, as well as large scale circulation variations described by AO or AAO.

The temporal evolution of ozone variations attributed to natural proxies and to EESC is presented in Figure 8. The figure shows time series of ozone anomalies and regression results from 1980 to 2015 SBUV monthly mean overpasses, averaged over 3 stations (Hohenpeißenberg, Haute Provence and Table Mountain). Two stratospheric layers are shown: Layer 8 (40.34-25.45 hPa) centered at about 24 km height and layer 13 (4.034-2.545 hPa) centered at about 40 km height. Figure 8 shows that the major long-term variations come from

chemistry (EESC), the solar cycle and AOD.

It appears however, that in some particular years the synergistic contribution of shorter-term variations can result in substantial additive anomalies. This might or might not influence the estimation of long term changes or trends in the ozone profile. Notable synergistic negative anomalies can be seen in the years 1983, 1985, 1988,

1992, 1993, 1995, 1997, 1999, 2002, 2004, 2006, 2008, 2011, 2013 in which the negative phase of QBO and of other proxies coincided. Further analysis, however, showed that, even after removing the above years, the observed long term variability / trends remained the same. We note here that the correlations between the regressed time series (all proxies composed) and the observed ozone anomalies are 0.62 for layer 8 (t-value = 16.19, p-value < 0.0001, N = 426) and 0.67 for layer 13 (t-value = 18.59, p-value < 0.0001, N = 426).






Another look at long-term ozone variations is given in Figure 9. The figure shows the observed SBUV overpass anomalies, the variations explained by natural influences (all proxies except EESC), and the remaining ozone variation after all natural influences and EESC (all proxies) have been subtracted, again for the whole 36-year period (1980-2015) and layers 8 and 13. From Figures 9a and 9b one can clearly see that removing the effect of

natural variation proxies has little effect on the slow long-term variations and on ozone trends. Most of the long-term variation is congruent with the EESC proxy, especially in the upper (Layer 13) stratosphere. EESC is really the dominating proxy for ozone trends in the upper stratosphere. When the EESC related variations are removed as well (lower panels), only very small long-term variations remain. There is a slightly negative tendency in the lower stratosphere after around 2000 and a very small positive tendency after 2000 in the upper stratosphere.

After removing the variability attributed to all proxies and EESC, the nonparametric Mann-Kendall rank statistic trend test (Mitchell et al., 1966) was applied to the anomaly series. It was found that both in the upper and lower stratosphere the overall trends (1980-2015) were insignificant at the 99% confidence level.

**5 Stratospheric ozone trends before and after 1998**

Various authors (Newchurch, 2003; Reinsel et al., 2005; Zanis et al., 2006; Zerefos et al., 2012; Harris et al.,

2015; Solomon et al., 2016; Steinbrecht et al., 2017) provide evidence for a difference in ozone "trends" before and after the years 1996/1998. Using the MLR model described in paragraph 4.1 we have calculated linear trends, with and without including the various proxies listed in 4.1, for the SBUV zonal means and SBUV overpasses over the lidar stations. Trends were calculated using the PWLT method (1/1998 set as inflection point). In a separate run we used EESC to describe the ozone trends, and from that calculated the EESC related

ozone trends before and after 1998.

As a first step, we performed a base-line run, fitting only the two linear trend terms (denoted as T1 and T2 in paragraph 4.1) and the volcanic effect (AOD). This gives the pre- and post- 1998 trends in the upper row of Figure 10 (10a). Then we performed a run with the PWLT method accounting for the effects QBO, ENSO, solar

cycle, tropopause variability, AO/AAO and volcanic effects, and including the two linear trend terms T1 and T2 for the same inflection point. The resulting trends are displayed in Figure 10b (mid-row). Finally, we performed a run with all proxies, but using EESC instead of PWLT. The corresponding ozone trends before and after 1998, due to the fitted EESC, are presented in Figure 10c (bottom row).

Comparison of the trends presented in Figures 10a and 10b, both calculated using linear trend terms (PWLT) shows minor changes only. Clearly, introducing the different proxies has very little effect on the trends. The proxy that has the largest influence on trends is the solar cycle. Comparison between Figures 10b and 10c shows that for the pre-1998 period (left panels) trends are very similar (almost identical), regardless if a linear trend term (the pre-1998 part of the PWLT method) or EESC are used. For the post-1998 period (right panels), results

are not so close to each other, albeit both show clear positive ozone trends after 1998.

While trends calculated with the use of EESC reflect the prescribed shape of the EESC curve, PWLT linear trends can react to other long-term changes, e.g. to effects of increasing Green House Gases (GHGs) and global



warming. Chemistry-Climate model simulations assessing the effects of changes in ODS and / or GHGs indicate

that their contributions add linearly to produce the overall ozone change (see detailed discussion and references in WMO, 2014, par. 2.3.5.2). This may explain some of the differences between panels 10b and 10c. However, a detailed investigation is beyond the scope of the present paper.

Although the period (1998-2015) is slightly larger from the period studied by Frith et al., 2017 (2001-2015) the

results reported here are in general agreement with the SBUV trends reported in that study. Finally, it should be noted that the profiles of trends from SBUV station overpasses (dashed lines) and trends for the latitudinal belts (solid lines) are very similar for both periods of study (1980-1997 and 1998-2015).

Figure 11 extends the previous findings to a global perspective, based on the SBUV zonal means. All cross

sections in Figure 11 are plotted against the sine of latitude north and south. The tick marks of the vertical axis are centered at the indicated pressure level. The color scale gives the calculated trends in percent per decade. The first vertical group of cross sections refers to the period 1980-2015, the middle to the period 1980-1997 and the right to the period 1998-2015. Comparing the observed trends during the different periods, we see that there is a region between 10 and 5 hPa over the tropics which shows positive ozone trends over the whole 1980 to 2015

period of record, and to a different degree also in the two sub-periods. Also notable are the negative trends over middle and high latitudes below 15 hPa, both in the total 1980 to 2015 period and in the sub-periods (except for the post 1998 period in EESC fit, lowermost right panel). The big change when dividing the 1980-2015 period into two sub-periods is the change in sign of the observed trends in the upper stratosphere, as well as in parts of the middle stratosphere, particularly over middle and high latitudes (upper set of cross sections).


The middle and lower sets of cross sections in Figure 11 are plotted to provide preliminary answers to the effect of including natural proxies, and to the congruence between PWLT and trends using the prescribed EESC curve. It is obvious from the top and middle panels of Figure 11 that adding or removing the natural proxies has little effect on the observed trends. The general similarity between the middle and bottom set of cross sections in

Figure 11 points out the importance of man-made ozone depleting substances, represented by EESC, for the observed (PWLT) trends, especially for the first sub-period 1980-1997. That similarity is not so clear when comparing cross sections during the last period 1998-2015. Here, EESC provides different patterns of upward trends than PWLT. In particular the EESC based trends after 1998 do not show the continuing downward trends seen in the lower stratosphere in the middle panels. The reasons must be quite complex. As already mentioned

above, one reason might be the continuing increase of GHGs and the general variability of the middle and lower stratosphere (e.g. Jonsson et al., 2004; Zerefos et al., 2014).

## 6 Conclusions

The paper investigates the representativeness of single lidar stations for our understanding of trends in the vertical ozone profile. From the lower to the upper stratosphere single or grouped stations correlate well with

zonal means calculated from SBUV overpasses. Good correlation (> 0.4) with zonal means is found within a few degrees of latitude north or south of any lidar station. The latitude range of representativeness expands as we move to the upper stratosphere, and spans a large part of the globe above 4 hPa. Ozone trend profiles are very





similar over the different stations and their corresponding zonal means. Detailed analysis of proxy footprints in the vertical ozone profiles also shows large similarities between lidar time series at the stations, SBUV overpass time series, and SBUV zonal means. Ozone trends have been studied with and without the inclusion of additional proxies, and for the full period 1980-2015, as well as for the two sub-periods 1980-1997 and 1998-2015. The major contributions to the trends come from man-made ozone depleting substances (EESC), the solar cycle and AOD. Long-term trends were not influenced by adding all other proxies, although these can produce significant negative anomalies in certain years, for example in 1983, 1985, 1988, 1992, 1993, 1995, 1997, 1999, 2002, 2004, 2006, 2008, 2011, 2013. During all these years ozone at about 24 km dropped below -6% of the mean.

A so-called "inflection point" between 1997 and 1999 marks the change from previously significant negative ozone trends, to recent positive ozone trends (1998-2015). This trend-change is observed at levels above 15 hPa, but is not always significant. Ozone trends in two periods before and after 1998 have been further compared with a multiple regression model with piece-wise linear trends (PWLT), with and without natural proxies, or with EESC representing the effects of man-made ozone depleting substances (ODS). Natural proxies had little effect on the observed trends in both periods before and after 1998. The largest contributor to the observed ozone trends in the 1980 to 1997 period were man-made ODS. In the second period 1998-2015, the decline of ODS / EESC is reflected in generally increasing PWLT trends. However, differences between latitude / altitude cross sections of post 1998 trends from PWLT versus EESC indicate that other influences are operating as well. There are long-term ozone changes that may have been caused by factors such as increasing GHGs.

Our analysis indicates that natural proxies in synergy (or not) cannot explain the different sign of ozone trends before and after the year 1998. Finally we note that we have plotted some figures with the sine of latitude in order to emphasize the tropics (which occupy about half of the globe). Latitude-altitude cross sections and analysis like the one presented here can provide clues for a better understanding of the changed trends of stratospheric ozone in the past decade or so. While declining EESC plays an important role, our findings indicate that other influences are operating as well. A pre-requisite for understanding these influences is to maintain NDACC and similar observing networks.

## 7 Data availability

Satellite SBUV ozone data overpassing Hohenpeißenberg, Haute Provence, Table Mountain, Mauna Loa and Lauder were obtained from ftp://toms.gsfc.nasa.gov/pub/sbuv/AGGREGATED/. Additional SBUV data at 5 degree of latitude zonal means were taken from ftp://toms.gsfc.nasa.gov/pub/MergedOzoneData/Ind_Inst_HDF/. Ground-based lidar ozone profiles were obtained from the NDACC Database at ftp://ftp.cpc.ncep.noaa.gov/ndacc/station/.

## Acknowledgements

The authors acknowledge the Mariolopoulos-Kanaginis Foundation for the Environmental Sciences for funding this research. We acknowledge the individual agencies and scientists responsible for the ground-based lidar





ozone measurements, particularly Thierry Leblanc for the Mauna Loa and Table Mountain lidars, and Daan
       Swart and Ann van Gijsel for the Lauder lidar, as well as the SBUV science team for providing the satellite
       ozone profiles. B.H. would like to thank DLR-IPA in Oberpfaffenhofen/Germany for funding a three month stay
       as visiting scientist that allowed her to work on this study.

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



**Table 1. Pressure layers in which ozone data have been analysed in this study.**

| Layer 8 | 40.34 - 25.45 hPa |
|---|---|
| Layer 9 | 25.45 - 16.06 hPa |
| Layer 10 | 16.06 - 0.13 hPa |
| Layer 11 | 10.13 - 6.393 hPa |
| Layer 12 | 6.393 - 4.034 hPa |
| Layer 13 | 4.034 - 2.545 hPa |
| Layer 14 | 2.545 - 1.606 hPa |


**Table 2. SBUV satellite ozone data coverage used in this study.**

| Nimbus 7 SBUV | 11/1978 - 05/1990 |
|---|---|
| NOAA-9 SBUV/2 | 02/1985 - 01/1998 |
| NOAA-11 SBUV/2 | 01/1989 - 03/2001 |
| NOAA-14 SBUV/2 | 03/1995 - 09/2006 |
| NOAA-16 SBUV/2 | 10/2000 - 05/2014 |
| NOAA-17 SBUV/2 | 08/2002 - 03/2013 |
| NOAA-18 SBUV/2 | 07/2005 - 11/2012 |
| NOAA-19 SBUV/2 | 03/2009 - present |






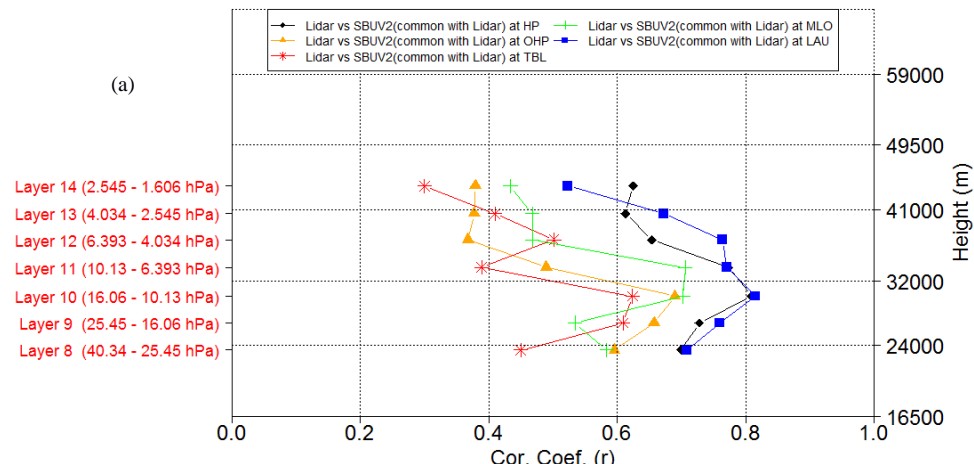

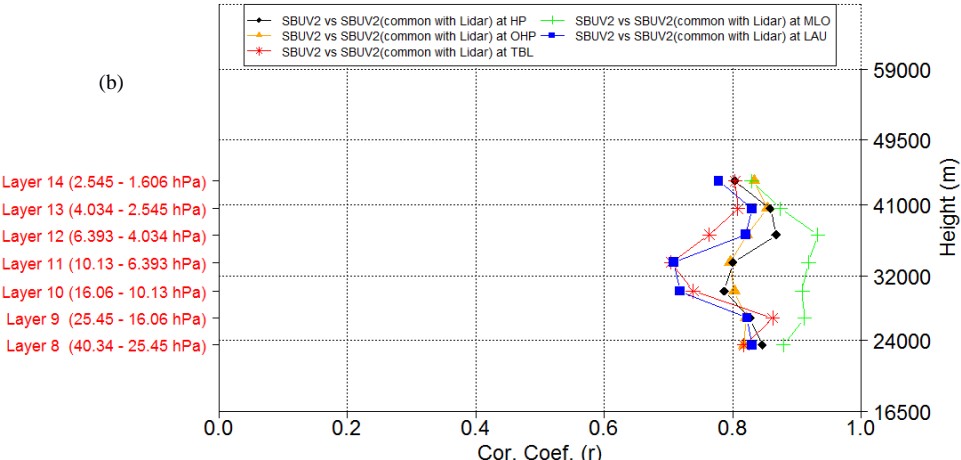

**Figure 1. (a) Correlation between monthly mean ozone anomalies from lidar and SBUV station overpasses on common days. Best correlations are between 25 and 32 km. All correlations are statistically significant at 99.99%. HP: Hohenpeißenberg, OHP: Haute Provence, TBL: Table Mountain, MLO: Mauna Loa, LAU: Lauder. (b) Same as in (a) but comparing monthly mean SBUV overpasses from about 30 days in a month with monthly mean SBUV overpasses from only days when lidar measurements were available. All correlations are statistically significant at 99.99%.**





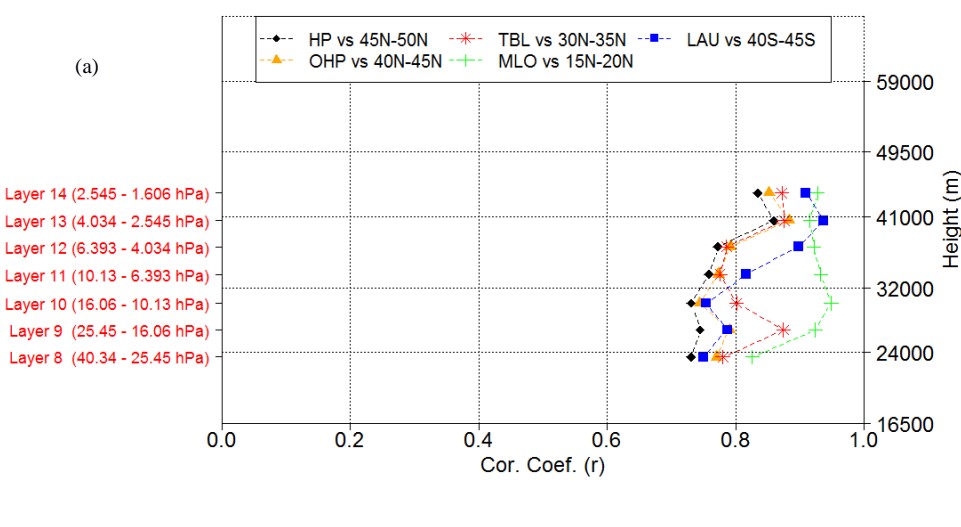

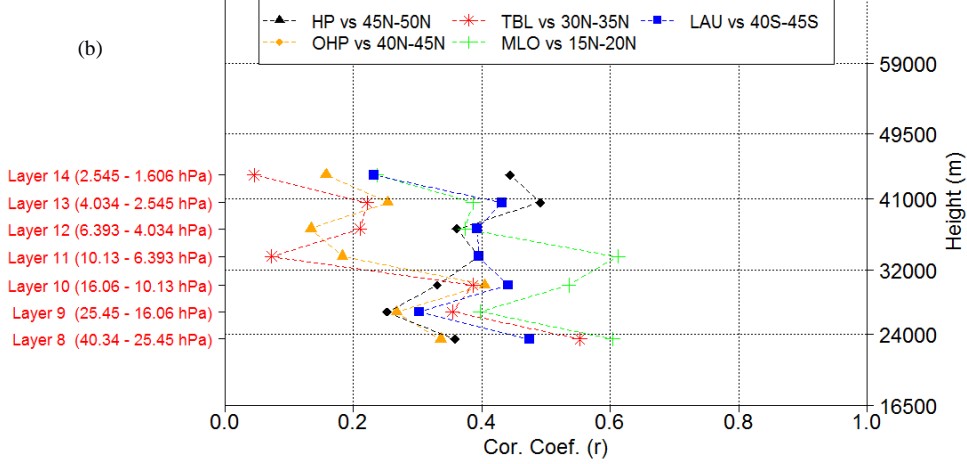

**Figure 2. (a) Correlations between monthly mean SBUV station overpasses and the corresponding SBUV monthly 5° zonal means. (b) Correlations between monthly mean lidar observations and the corresponding SBUV monthly 5° zonal means.**



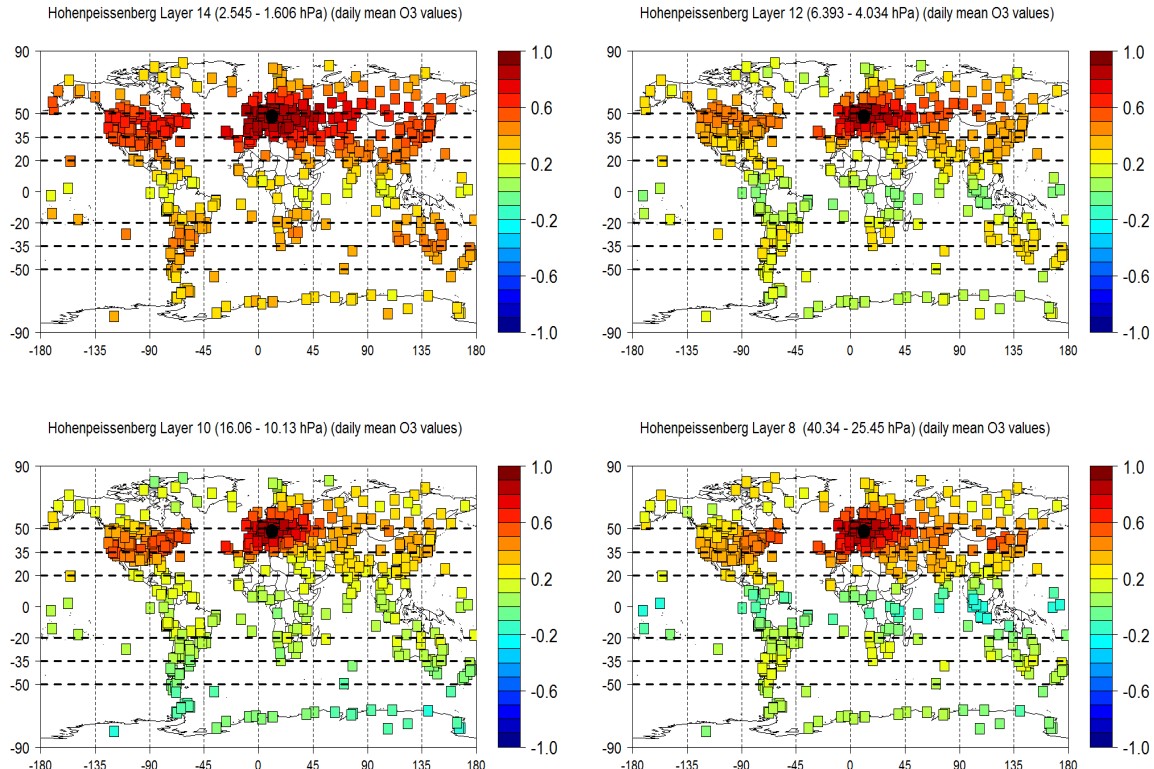

**Figure 3. Spatial distribution of correlations between monthly mean SBUV overpasses at Hohenpeißenberg and monthly mean SBUV overpasses at all other available stations. Correlations are given for four selected layers. Black dots indicate the location of Hohenpeißenberg.**



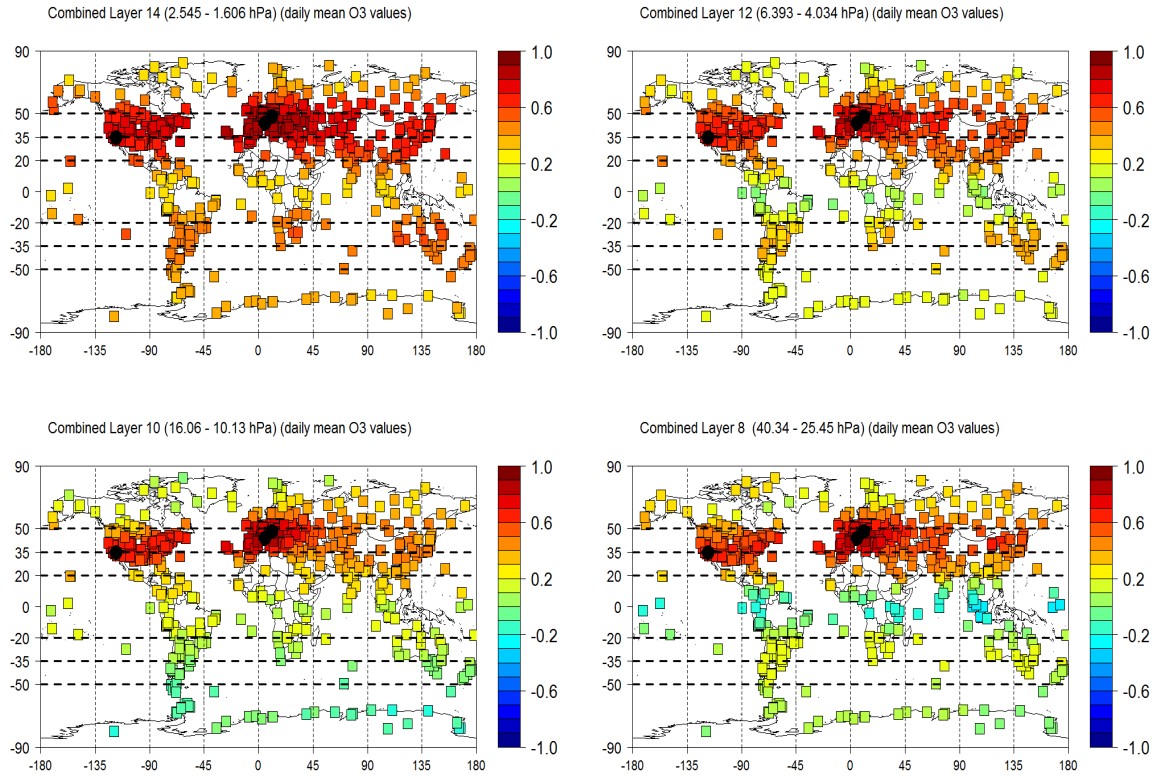

**Figure 4. Same as in Figure 3 but for three combined northern mid-latitude stations (Hohenpeißenberg, Haute Provence and Table Mountain). Black dots indicate the locations of Hohenpeißenberg, Haute Provence and Table Mountain.**



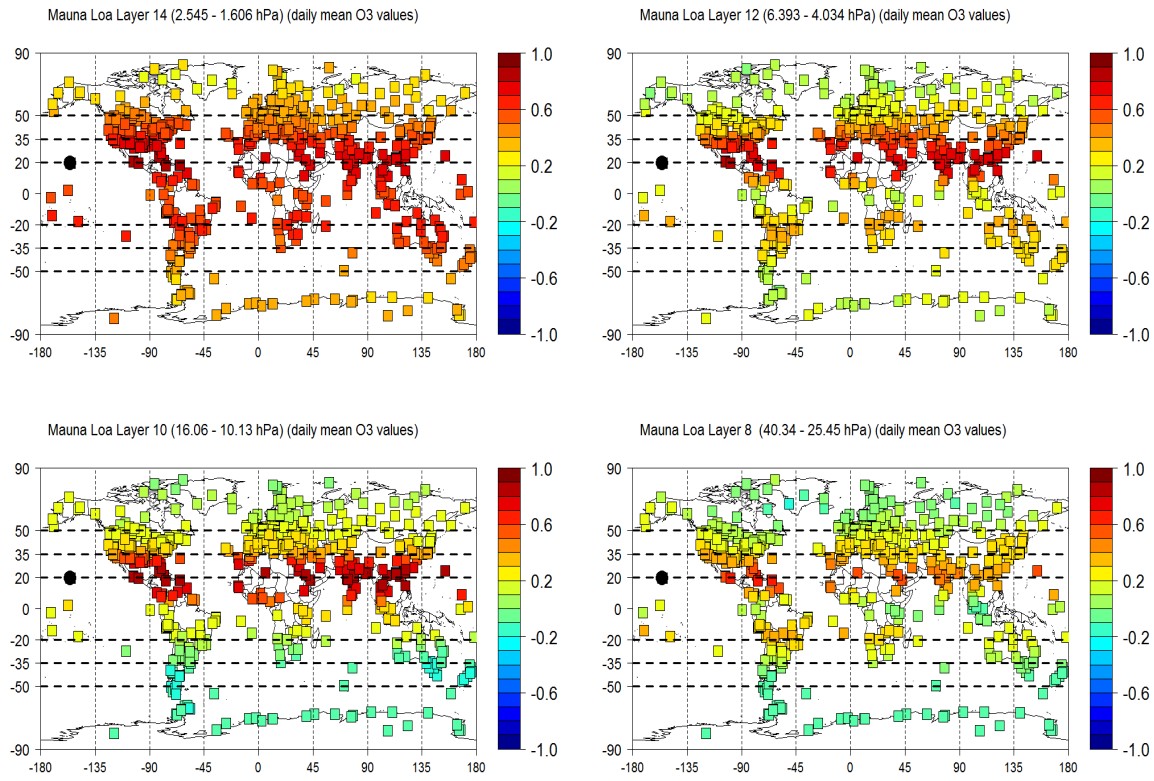

Figure 5. Same as in Figure 3 but for Mauna Loa. Black dots indicate the location of Mauna Loa.



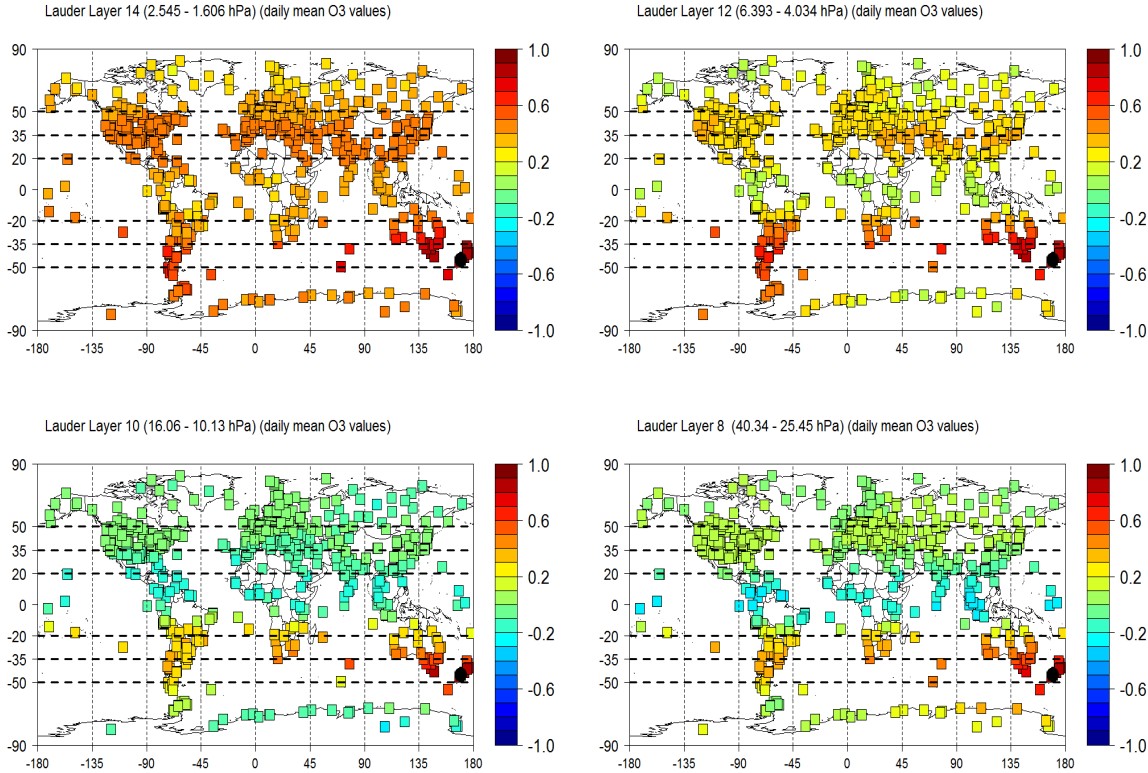

**Figure 6. Same as in Figure 3 but for Lauder. Black dots indicate the location of Lauder.**





**Figure 7. Amplitudes [i.e. (max-min) / 2] of ozone variations attributed to EESC, QBO, F10.7, MEI, Tropopause pressure, AO (or AAO at Lauder) and AOD for each stratospheric layer. All values are expressed in % of the long-term mean at each layer. Stations shown are: Hohenpeißenberg (47.8° N, 11.0° E), Mauna Loa (19.5° N, 155.6° W) and Lauder (45.0° S, 169.7° E). Left panel: SBUV overpass data for the full period 1980-2015. Middle panel: SBUV overpass data for common period with lidar, starting in 1987 at Hohenpeißenberg, 1993 at Mauna Loa, and 1994 in Lauder. Right panel: lidar monthly means.**





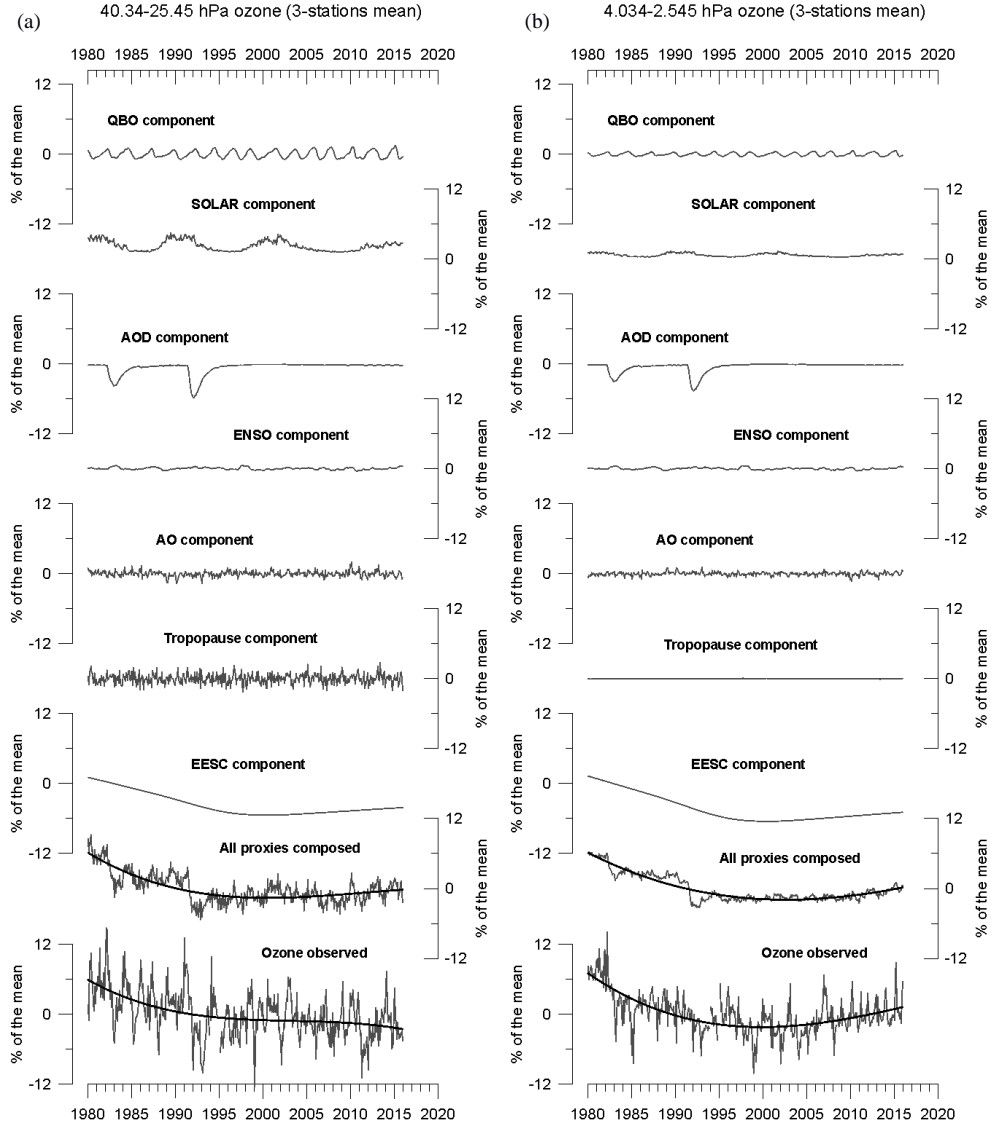

**Figure 8. (a) Ozone variations attributed to the different proxies (QBO, Solar, AOD, ENSO, AO, Tropopause, EESC) at layer 8 (40.34-25.45 hPa, centered at about 24 km height) for SBUV overpasses averaged over Hohenpeißenberg, Haute Provence and Table Mountain. (b) Same as in (a) but for layer 13 (4.034-2.545 hPa) centered at about 40 km height. The lower most curves give the observed deseasonalized SBUV time series. Thick solid curves in the four bottom panels are third degree polynomials fit to the data.**



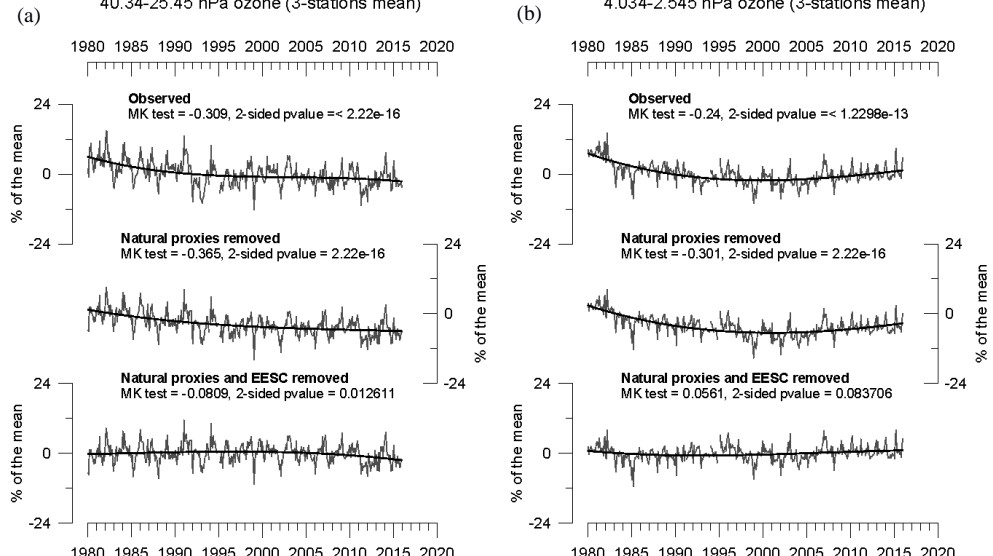

**Figure 9.** Ozone anomalies from SBUV overpasses averaged over Hohenpeißenberg, Haute Provence and Table Mountain. Top: Original deseasonalized time series. Middle: Time series with natural proxies removed, but EESC related variations remaining. Bottom: Time series with natural proxies and EESC related variations removed. (a) For layer 8 (40.34-25.45 hPa, centered at about 24 km height). (b) Same as in (a) but for the layer 13 (4.034-2.545 hPa) centered at about 40 km height. MK test refers to the Mann-Kendall trend test. Thick solid curves are third degree polynomials fit to the data.




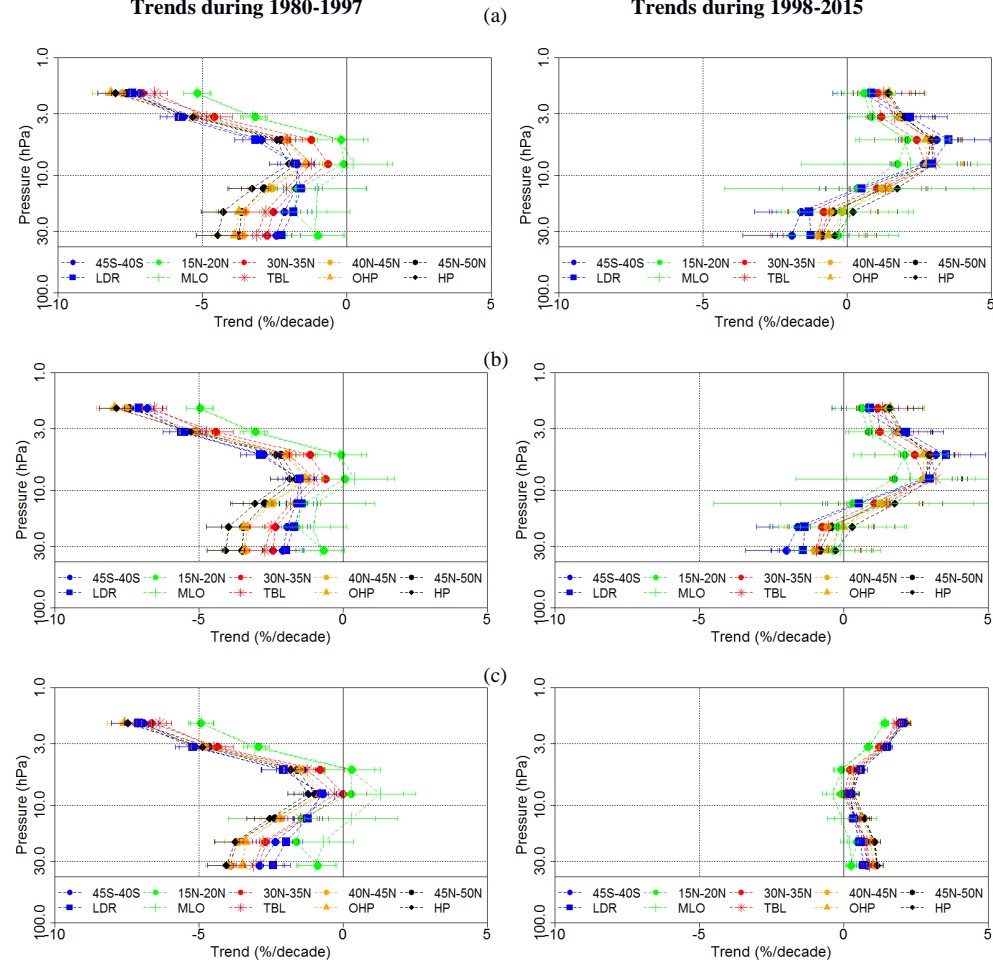

**Figure 10. Trends in the vertical distribution of ozone for the pre-1998 and post-1998 period, based on SBUV station overpass and zonal mean data, using (a) two linear trend terms and volcanic effects only, (b) the PWLT method including all proxies, and (c) using all proxies and EESC to describe the long term ozone changes.**







**Figure 11.** Cross-section of ozone trends from zonal mean SBUV (1980-2015, left), (1980-1997, middle) and (1998-2015, right) in percent per decade. Rows as in Figure 10, upper: PWLT, no proxies except AOD, middle: PWLT with all proxies, bottom: trends from fitted EESC. Stippling indicate significance at 95%.