# Peer review of "Representativeness of single lidar stations for zonally averaged ozone profiles, their trends and attribution to proxies"

_Atmospheric Chemistry and Physics, 2017_

## Referee Comment (RC1) · Anonymous Referee #1 · 3 Feb 2018

This paper compares SBUV and lidar ozone data from multiple stations to assess the representativeness of the lidar data of the global variability at different vertical regions. Additionally, it makes use of this data and a MLR analysis using various combinations of different proxies to assess the influence of these proxies on stratospheric ozone trends, which is important in the current context of ongoing work to determine the status of ozone recovery in the stratosphere. The analysis techniques seem thorough and comprehensive, although some additional explanation may be necessary for the purpose of clarification. Additionally, the results seem sound though the authors may overlook some more simple conclusions about the limitations of the methodology. I would recommend this paper for publication subject to some minor revisions detailed

below.

**Major Comments**

Pg. 07, Ln. 247: "Important aspects of Figure 7 are . . ."

I agree that the total amount of variability is similar between the lidars and SBUV over the lidar time period for most of the stratosphere but it may be worth noting and explaining the discrepancies at the highest altitudes (i.e., lidar data quality diminishes) and lowest altitudes (i.e., SBUV data quality and resolution diminishes). Additionally there appears to be poorer agreement for Lauder than for the other stations.

While, for many comparisons, the total variability is similar the individual attributions for different proxies can be very different. For example, Hohenpeissenberg shows systematically larger EESC responses than SBUV over the same time period at all altitudes. Another example is how the AOD responses can be very different across all figures (except at Hohenpeissenberg), making one question if the regressions have sufficient ability to separate the influence of volcanic aerosol from other proxies given different time periods.

Pg. 08, Ln. 281: "It was found that both in the upper and lower stratosphere the overall trends (1980-2015) were insignificant at the 99% confidence level."

It is important to remember the limitations of these regression analyses. It is generally highly unlikely that, after performing a MLR analysis to the data (i.e., a least-squares fitting technique), the residuals will have any trend-like behavior. However, that only means that on the whole the data is represented, not that the individual attributions to the different proxies are correct. For example, most MLR analyses to ozone cited by this manuscript (and apparently this manuscript's data as well) show negative monotonic trends in the lower stratosphere. These are expected to be primarily the result of the influence of greenhouse gases on increasing the strength of the Brewer Dobson circulation. What this means is that using the EESC as a proxy here, which

forces a turnaround, is incorrect. Please see Kuttippurath et al., GRL, 2015 (DOI: 10.1002/2014GL062142) for more details. This is the main explanation for the nature of the residuals shown in Figure 9a at the bottom. It also means that the overall representation for the influence of the EESC on ozone in the lower stratosphere is incorrect in Figure 7. This is, of course, clearly evident in Figure 10b. In order to use incorporate the EESC in a way that allows for monotonic trends, you would need to use multiple EESC proxies such as those introduced in Damadeo et al., ACP, 2014 (DOI: 10.5194/acp-14-13455-2014).

Figure 10: It seems strange that the error bars in 10b and 10c in the left panel are comparable but are much smaller in 10c than 10b in the right panel. Do you have an explanation for how the uncertainties in these small recovery trends shrank so drastically?

**Minor Comments**

Pg. 03, Ln. 092: "Figure 1a shows the resulting correlation coefficients. All correlations are statistically significant at the 99.99% confidence level."

What kind of correlation coefficient is this (e.g., the Pearson product-moment correlation) and how was the statistical significance computed?

Can you elaborate more on what data is being correlated in figures 1 and 2? For example, figure 1a states it is plotting "correlation between monthly mean ozone anomalies from lidar and SBUV station overpasses on common days." Does this mean that you are computing SBUV overpasses over the lidar stations and then taking monthly means of those? If so, what are the coincident criteria? Or are you doing something different. It isn't clear how the data is being processed or binned for these different correlation comparisons.

Pg. 04, Ln. 140: "The fairly good "zonal representativeness" of the stations is obvious ..."

It is also worth noting the obvious patterns in the correlations. The data being used here are deseasonalized anomalies, so correlations are expected to be larger in the presence of larger geophysical variability that is represented in multiple zonal regions. For example, stations at midlatitudes show worse correlations with data in the tropics at lower altitudes where the QBO has a much larger amplitude. Similarly, stations at midlatitudes show better correlations at midlatitudes in the opposite hemisphere at the highest altitudes because the long-term trend is a significant source of variability here but has more similar trend values and turnaround times at midlatitudes than in the tropics. The nature of resulting patterns in solar cycle amplitude as they pertain to ozone variability in the middle to upper stratosphere also play a role.

Pg. 07, Ln. 264: "Notable synergistic negative anomalies can be seen . . ."

Why would you consider these anomalies if they are also represented in the data? This is perfectly acceptable as part of the regression analysis and these years should not be ignored. If the coincidental phasing of proxies was not represented by the data, then that would be considered an outlier or anomaly and would need to be considered.

Pg. 08, Ln. 292: "As a first step . . ."

Did you also test just using the PWLT to compare the influence of adding the AOD proxy?

Pg. 08, Ln. 301: "The proxy that has the largest influence on trends is the solar cycle."

It should be noted that the solar cycle can have a larger influence on these MLR analyses if the data record being utilized is smaller (i.e., less than 2 solar cycles).

Pg. 08, Ln. 304: "For the post-1998 period (right panels), results are not so close to each other, albeit both show clear positive ozone trends after 1998."

Both show positive ozone trends after 1998 above about 15 hPa. Of course, for reasons stated earlier, the EESC proxy are forced to show positive trends during this time period regardless of whether the trends are actually positive.
Pg. 09, Ln. 323: "... we see that there is a region between 10 and 5 hPa over the tropics which shows positive ozone trends over the whole 1980 to 2015 period of record ..."

Although these are not statistically significant.

Pg. 09, Ln. 333: "It is obvious from the top and middle panels of Figure 11 that adding or removing the natural proxies has little effect on the observed trends."

Except for the AOD proxy?

Pg. 09, Ln. 339: "The reasons must be quite complex."

Not really. As mentioned earlier, they simply can't from a mathematical standpoint.

Figure 11: What sort of zonal binning scale was used here?

---

## Referee Comment (RC2) · Anonymous Referee #2 · 13 Feb 2018

This paper uses correlations and trends to examine the similarities between ground-based lidar measurements of ozone profiles with SBUV observations in order to assess the representativeness of the ground stations of zonal and global behaviour. The work is interesting and suitable for publication in ACP; however, as pointed out by the other reviewer, several topics require more detailed explanation. As well, it seems that work could provide a more quantitative assessment of the representativeness of the ground stations. As it stands, the conclusions are vague. I would challenge the authors to consider pushing the analysis to provide better quantification of this representativeness. For example the correlations in Figs 3-6 are interesting and could be presented in a more condensed fashion that would provide some actual numbers about the geophys-

ical behaviour. In addition to this, the following minor comments should be addressed.

Section 3: It is not clear how the reduced vertical structure in the correlation for monthly means shows that the decreased correlation above 35 is due to instrumental differences. Purely random variability would also average out in the means and increase the correlation. As requested by the other reviewer, please provide details on how all these correlation calculations are performed so that the analysis is repeatable. Also, the role of atmospheric variability is important here and should be discussed in detail, e.g. the strength of the QBO in altitude and latitude has a very strong and predictable effect on correlations.

Section 4.1: How is the tropopause pressure term "filtered"? Also, the GloSSAC data set would be a much better choice for the AOD and would not require artificial extension of the end of the ozone data record (Thomason, et al.: A global, space-based stratospheric aerosol climatology: 1979 to 2016, Earth Syst. Sci. Data Discuss., https://doi.org/10.5194/essd-2017-91, 2017.)

Section 4.2: The large difference in contribution from AOD between lidars and SBUV at the highest altitudes should be noted and explained if possible. This sweeping statement (on line 250 page 7) "Together with the correlations shown in the previous section, this means that ground-based instruments at single stations can provide representative information about ozone trends (EESC) and ozone variations related to the QBO, the solar cycle, ENSO, as well as large scale circulation variations described by AO or AAO" really needs quantification, i.e. to what degree, what "information", over what scale? And on line 260, the major contribution from AOD is highly limited to the two strong eruption time periods.

---

## Author Comment (AC1) · 27 Mar 2018

**Reply to Reviewer #1**

The authors would like to thank Reviewer #1 for his constructive comments and suggestions.

**Major Comments**

**Major comment 1**

Pg. 07, Ln. 247: "Important aspects of Figure 7 are ..."

I agree that the total amount of variability is similar between the lidars and SBUV over the lidar time period for most of the stratosphere but it may be worth noting and explaining the discrepancies at the highest altitudes (i.e., lidar data quality diminishes) and lowest altitudes (i.e., SBUV data quality and resolution diminishes). Additionally there appears to be poorer agreement for Lauder than for the other stations.

While, for many comparisons, the total variability is similar the individual attributions for different proxies can be very different. For example, Hohenpeissenberg shows systematically larger EESC responses than SBUV over the same time period at all altitudes. Another example is how the AOD responses can be very different across all figures (except at Hohenpeissenberg), making one question if the regressions have sufficient ability to separate the influence of volcanic aerosol from other proxies given different time periods.

**Answer to major comment 1:**

Figure 7 (now Figure 4) has been completely revised and the new results are now based on two orthogonal EESC terms as suggested by Reviewer 1. It was found that the use of 2 orthogonal EESC terms in the regression model improves the comparisons between the lidar and the SBUV data in the new figure. "We have confined our analysis to the SBUV layers from 8 (40-25 hPa) up to 14 (2.5-1.6 hPa). This was imposed by the fact that at the highest altitudes lidar data quality is reduced while at the lowest altitudes SBUV data quality is also reduced"; this has been added in section 3. An additional improvement arose from the use of AOD as suggested by Reviewer # 2. We have used throughout this paper the AOD records provided by Thomason et al. (2018).

The text on page 7 has been revised and now reads as follows:

"As it appears from Figure 4 the percent of the total variability explained by all proxies taken together, ranges between 5 and 15% of the mean, both for the lidar and the SBUV overpasses. Additionally there appears to be poorer agreement for Lauder than for the other stations. It should be noted here that both Lauder and Mauna Loa lidar records start 1-2 years after the Mt. Pinatubo eruption and this makes it difficult to separate the influence of volcanic aerosol from other proxies at these two stations, but not at Hohenpeissenberg because of its longer record."

**Major comment 2**

*Pg.* 08, *Ln.* 281: "It was found that both in the upper and lower stratosphere the overall trends (1980-2015) were insignificant at the 99% confidence level."

It is important to remember the limitations of these regression analyses. It is generally highly unlikely that, after performing a MLR analysis to the data (i.e., a least squares fitting technique), the residuals will have any trend-like behavior. However, that only means that on the whole the data is represented, not that the individual attributions to the different proxies are correct. For example, most MLR analyses to ozone cited by this manuscript (and apparently this manuscript's data as well) show negative monotonic trends in the lower stratosphere. These are expected to be primarily the result of the influence of greenhouse gases on increasing the strength of the Brewer Dobson circulation. What this means is that using the EESC as a proxy here, which forces a turnaround, is incorrect. Please see Kuttippurath et al., GRL, 2015 (DOI:10.1002/2014GL062142) for more details. This is the main explanation for the nature of the residuals shown in Figure 9a at the bottom. It also means that the overall representation for the influence of the EESC on ozone in the lower stratosphere is incorrect in Figure 7. This is, of course, clearly evident in Figure 10b. In order to incorporate the EESC in a way that allows for monotonic trends, you would need to use multiple EESC proxies such as those introduced in Damadeo et al., ACP, 2014 (DOI:10.5194/acp-14-13455-2014).

**Answer to major comment 2:**

We would like to thank the reviewer for pointing out the correct use of the EESC proxy. In the revised manuscript we incorporate the EESC proxy as suggested and now use two EESC terms (EESC and its orthogonal term) as introduced in Damadeo et al. (2014). The Figures 7, 8, 9, 10, 11 (now Figures 4, 5, 6, 7, 8) and Section 4.1 (Regression analysis model) have been revised accordingly. The references Kuttippurath et al., GRL, 2015 (DOI:10.1002/2014GL062142) and Damadeo et al., ACP, 2014 (DOI:10.5194/acp-14-13455-2014) have been added to the list of references.

**Major comment 3**

Figure 10: It seems strange that the error bars in 10b and 10c in the left panel are comparable but are much smaller in 10c than 10b in the right panel. Do you have an explanation for how the uncertainties in these small recovery trends shrank so drastically?

**Answer to major comment 3:**

On Fig. 10c (right panel) the reviewer was right because we found out a plotting error which is now corrected. The Figure 10 (now Figure 7) has been revised.

**Minor Comments**

**Minor comment 1**

*Pg.* 03, *Ln.* 092: "Figure 1a shows the resulting correlation coefficients. All correlations are statistically significant at the 99.99% confidence level."

What kind of correlation coefficient is this (e.g., the Pearson product-moment correlation) and how was the statistical significance computed?

Can you elaborate more on what data is being correlated in figures 1 and 2? For example, figure 1a states it is plotting "correlation between monthly mean ozone anomalies from lidar and SBUV station overpasses on common days." Does this mean that you are computing SBUV overpasses over the lidar stations and then taking monthly means of those? If so, what are the coincident criteria? Or are you doing

something different. It isn't clear how the data is being processed or binned for these different correlation comparisons.

Answer to 1: The following text was added for clarification in section 3.

"The comparison between lidar and SBUV station overpasses on common days throughout the record was based on deseasonalized monthly mean lidar and SBUV ozone profiles. Figure 1a shows the resulting correlation coefficients which were found to be all statistically significant at the 99.99% confidence level. Concerning the correlations between lidar data and SBUV overpasses, we calculated monthly averages when at least 3 common days were available. The data were deseasonalized by subtracting the long-term monthy mean pertaining to the same calendar month. All correlation coefficients (r) were calculated using the Pearson product-moment correlation and were tested for significance using the t-test formula for the correlation coefficient with n-2 degrees of freedom (von Storch and Zwiers, 1999):

$$t = r\sqrt{\frac{n-2}{1-r^2}} \tag{2}$$

We have added the reference in the list of references:

"von Storch, H., and Zwiers, F. W.: Statistical analysis in climate research, Cambridge University Press, Cambridge, 484 pp, ISBN 0 521 45071 3, 1999."

**Minor comment 2**

Pg. 04, Ln. 140: "The fairly good "zonal representativeness" of the stations is obvious ..."

It is also worth noting the obvious patterns in the correlations. The data being used here are deseasonalized anomalies, so correlations are expected to be larger in the presence of larger geophysical variability that is represented in multiple zonal regions. For example, stations at midlatitudes show worse correlations with data in the tropics at lower altitudes where the QBO has a much larger amplitude. Similarly, stations at midlatitudes show better correlations at midlatitudes in the opposite hemisphere at the highest altitudes because the long-term trend is a significant source of variability here but has more similar trend values and turnaround times at midlatitudes than in the tropics. The nature of resulting patterns in solar cycle amplitude as they pertain to ozone variability in the middle to upper stratosphere also play a role.

Answer to 2: We have fully revised Figures 3-6 (now Figure 3) which now present the correlations in a more condensed way as requested also by Reviewer #2.

The last paragraph of section 3 describing old Figs 3-6 has been corrected and now reads as follows: "A further look at the spatial distribution of correlation coefficients between single SBUV overpasses at lidar stations (or station groups) and SBUV 5° zonal means is given in Figure 3. The correlation coefficients have been calculated using deseasonalized and detrended ozone data. Data were detrended by removing a 2-degree polynomial fit from the deseasonalized time series. The results show that ozone at the five selected lidar stations correlate well with ozone over a fairly wide range of latitudes within  $\pm$  15 degrees centered at the station. This result has little dependence on height. The correlation coefficients found were high and in all cases their statistical significance exceeded 99.99% (correlations ranging between 0.45 and

0.9 with the highest values near the latitude circle corresponding to each station). The fairly good "zonal representativeness" of the stations is obvious from the colour scale. We remind here that long-term trends have been removed from the time series and therefore long-term trends do not contribute to the observed correlations. As to the role played by other sources of natural variability e.g. QBO, solar cycle etc., which were not removed from the time series in the correlations, it is small and does not exceed 2-4% of the explained variance as seen in a separate analysis (not shown here)."

**Minor comment 3**

Pg. 07, Ln. 264: "Notable synergistic negative anomalies can be seen ... "

Why would you consider these anomalies if they are also represented in the data? This is perfectly acceptable as part of the regression analysis and these years should not be ignored. If the coincidental phasing of proxies was not represented by the data, then that would be considered an outlier or anomaly and would need to be considered.

Answer to 3: We thought worthwhile presenting a list of years with coincidental phase of anomalies of various proxies and to stress the fact that "synergistic" effect by different proxies has not influenced the trend.

The text has been revised accordingly.

**Minor comment 4**

Pg. 08, Ln. 292: "As a first step ..."

Did you also test just using the PWLT to compare the influence of adding the AOD proxy?

Answer to 4: Yes we tested with and without the AOD proxy and found insignificant difference on the trends. See also answer to minor comment 8.

**Minor comment 5**

*Pg.* 08, *Ln.* 301: "The proxy that has the largest influence on trends is the solar cycle."

It should be noted that the solar cycle can have a larger influence on these MLR analyses if the data record being utilized is smaller (i.e., less than 2 solar cycles).

Answer to 5: The sentence has been corrected and now reads as follows: "The proxy that has the largest influence on trends is the solar cycle, a result based on 36 years of data."

**Minor comment 6**

Pg. 08, Ln. 304: "For the post-1998 period (right panels), results are not so close to each other, albeit both show clear positive ozone trends after 1998."

Both show positive ozone trends after 1998 above about 15 hPa. Of course, for reasons stated earlier, the EESC proxy are forced to show positive trends during this time period regardless of whether the trends are actually positive.

Answer to 6: The use of 2 orthogonal EESC terms in the statistical regression model has improved the calculations of the post-1998 trends. The text has been corrected and now reads "For the post-1998 period (right panels), the resulting trends do not change significantly when comparing Figures 7a, 7b and 7c."

**Minor comment 7**

Pg. 09, Ln. 323: "... we see that there is a region between 10 and 5 hPa over the tropics which shows positive ozone trends over the whole 1980 to 2015 period of record ..."

Although these are not statistically significant.

Answer to 7: We note this and the new text reads now as follows: "Comparing the observed trends during the different periods, we see that there is a region between 10 and 5 hPa over the tropics which shows positive ozone trends over the whole 1980 to 2015 period of record, and to a different degree also in the two sub-periods. These trends however are not statistically significant."

**Minor comment 8**

*Pg. 09, Ln. 333: "It is obvious from the top and middle panels of Figure 11 that adding or removing the natural proxies has little effect on the observed trends." Except for the AOD proxy?*

Answer to 8: The AOD proxy has little and insignificant effect on the trends. The text has been corrected and now reads "It is obvious from the top and middle panels of Figure 8 that adding or removing the natural proxies has little effect on the observed trends. At any rate a separate analysis (not shown here) confirms that adding or removing of AOD has little and insignificant effect on the trends".

**Minor comment 9**

Pg. 09, Ln. 339: "The reasons must be quite complex." Not really. As mentioned earlier, they simply can't from a mathematical standpoint.

Answer to 9: To avoid confusion the statement "The reasons must be quite complex" has been removed.

Minor comment 10 Figure 11: What sort of zonal binning scale was used here?

Answer to 10: "Data are averaged over 5 degrees of latitude zones". This is added in the caption of the figure (now Figure 8).

**Reply to Reviewer #2**

We would like to thank Reviewer #2 for the constructive comments and suggestions.

**General comments**

This paper uses correlations and trends to examine the similarities between groundbased lidar measurements of ozone profiles with SBUV observations in order to assess the representativeness of the ground stations of zonal and global behaviour. The work is interesting and suitable for publication in ACP; however, as pointed out by the other reviewer, several topics require more detailed explanation. As well, it seems that work could provide a more quantitative assessment of the representativeness of the ground stations. As it stands, the conclusions are vague. I would challenge the authors to consider pushing the analysis to provide better quantification of this representativeness. For example the correlations in Figs 3-6 are interesting and could be presented in a more condensed fashion that would provide some actual numbers about the geophysical behaviour. In addition to this, the following minor comments should be addressed.

**Reply to general comments:**

We have fully revised Figures 3-6 (now Figure 3) which now present the correlations in a more condensed way as requested.

The last paragraph of section 3 describing old Figs 3-6 has been corrected and now reads as follows: "A further look at the spatial distribution of correlation coefficients between single SBUV overpasses at lidar stations (or station groups) and SBUV 5° zonal means is given in Figure 3. The correlation coefficients have been calculated using deseasonalized and detrended ozone data. Data were detrended by removing a 2-degree polynomial fit from the deseasonalized time series. The results show that ozone at the five selected lidar stations correlate well with ozone over a fairly wide range of latitudes within  $\pm$  15 degrees centered at the station. This result has little dependence on height. The correlation coefficients found were high and in all cases their statistical significance exceeded 99.99% (correlations ranging between 0.45 and 0.9 with the highest values near the latitude circle corresponding to each station). The fairly good "zonal representativeness" of the stations is obvious from the colour scale. We remind here that long-term trends have been removed from the time series and therefore long-term trends do not contribute to the observed correlations. As to the role played by other sources of natural variability e.g. QBO, solar cycle etc., which were not removed from the time series in the correlations, it is small and does not exceed 2-4% of the explained variance as seen in a separate analysis (not shown here)."

The conclusions and abstract have been revised.

**Minor comments**

**Minor comment 1**

Section 3: It is not clear how the reduced vertical structure in the correlation for monthly means shows that the decreased correlation above 35 is due to instrumental differences. Purely random variability would also average out in the means and

increase the correlation. As requested by the other reviewer, please provide details on how all these correlation calculations are performed so that the analysis is repeatable. Also, the role of atmospheric variability is important here and should be discussed in detail, e.g. the strength of the QBO in altitude and latitude has a very strong and predictable effect on correlations.

**Answer to 1:**

Please see the answers to major comment 1 and to minor comments 1 and 2 of Reviewer #1.

**Minor comment 2**

Section 4.1: How is the tropopause pressure term "filtered"? Also, the GloSSAC data set would be a much better choice for the AOD and would not require artificial extension of the end of the ozone data record (Thomason, et al.: A global, spacebased stratospheric aerosol climatology: 1979 to 2016, Earth Syst. Sci. Data Discuss., https://doi.org/10.5194/essd-2017-91, 2017.)

**Answer to 2:**

The tropopause pressure term was filtered by removing the natural oscillations (seasonal, QBO, ENSO, solar cycle, long-term trend and AOD) from the tropopause pressure data using multiple linear regression (MLR) analysis in the same way as we have done with the ozone data.

We have repeated our analysis with the AOD proxy from Thomason et al. (2018) as suggested by Reviewer #2. Section 4.1 has been revised accordingly.

**Minor comment 3**

Section 4.2: The large difference in contribution from AOD between lidars and SBUV at the highest altitudes should be noted and explained if possible. This sweeping statement (on line 250 page 7) "Together with the correlations shown in the previous section, this means that ground-based instruments at single stations can provide representative information about ozone trends (EESC) and ozone variations related to the QBO, the solar cycle, ENSO, as well as large scale circulation variations described by AO or AAO" really needs quantification, i.e. to what degree, what "information", over what scale? And on line 260, the major contribution from AOD is highly limited to the two strong eruption time periods.

**Answer to 3:**

Please see the answer to major comment 1 of Reviewer #1.

The sweeping statement has been removed.

The sentence "The major contribution from AOD is highly limited to the two periods with the strong volcanic eruptions (El Chichon and Pinatubo)" has been added in Section 4.2.

**Representativeness of single lidar stations for zonally averaged ozone profiles, their trends and attribution to proxies**

Christos Zerefos1,2, John Kapsomenakis1, Kostas Eleftheratos3, Kleareti Tourpali4, Irina Petropavlovskikh5, Daan Hubert6, Sophie Godin-Beekmann7, Wolfgang Steinbrecht8, Stacey 5 Frith9, Viktoria Sofieva10, Birgit Hassler11

[revised manuscript text omitted]

Field Code Changed

Field Code Changed

Field Code Changed

WMO Scientific Assessment of Ozone Depletion: 2006, Global Ozone Research and Monitoring Project-Report No. 50, WMO (World Meteorological Organization), Geneva, Switzerland, available at: https://www.esrl.noaa.gov/csd/assessments/ozone/, 2007.

- 540 WMO Scientific Assessment of Ozone Depletion: 2010, Global Ozone Research and Monitoring Project-Report No. 52, WMO (World Meteorological Organization), Geneva, Switzerland, available at: https://www.esrl.noaa.gov/csd/assessments/ozone/, 2011.
  - WMO Scientific Assessment of Ozone Depletion: 2014, Global Ozone Research and Monitoring Project-Report No. 55, WMO (World Meteorological Organization), Geneva, Switzerland, available at: https://www.esrl.noaa.gov/csd/assessments/ozone/, 2014.
  - Zanis, P., Maillard, E., Staehelin, J., Zerefos, C., Kosmidis, E., Tourpali, K., and Wohltmann, I.: On the turnaround of stratospheric ozone trends deduced from the reevaluated Umkehr record of Arosa, Switzerland, J. Geophys. Res., 111, D22307, https://doi.org/10.1029/2005JD006886, 2006.
  - Zerefos, C. S.: On the quasi-biennial oscillation in equatorial stratospheric temperatures and total ozone, Adv. Space Res., 2, 5, 177-181, 1983.
  - Zerefos, C. S., Bais, A. F., Ziomas, I. C., and Bojkov, R. D.: On the relative importance of quasi-biennial oscillation and El Nino/southern oscillation in the revised Dobson total ozone records, J. Geophys. Res., 97, D9, 10135-10144, 1992.
- Zerefos, C. S., Tourpali, K., Eleftheratos, K., Kazadzis, S., Meleti, C., Feister, U., Koskela, T., and Heikkila, A.:
  Evidence of a possible turning point in solar UV-B over Canada, Europe and Japan, Atmospheric Chemistry and Physics, 12, 2469–2477, doi: 10.5194/acp-12-2469-2012, 2012.
  - Zerefos, C. S., Tourpali, K., Zanis, P., Eleftheratos, K., Repapis, C., Goodman, A., Wuebbles, D., Isaksen, I. S. A., and Luterbacher, J.: Evidence for an earlier greenhouse cooling effect in the stratosphere before 1980 over the Northern Hemisphere, Atmospheric Chemistry and Physics, 14, 7705–7720, doi: 10.5194/acp-14-7705-2014, 2014.

Field Code Changed

Field Code Changed

**Field Code Changed**

545

550

Table 1. Pressure layers in which ozone data have been analysed in this study.

| Layer 8  | 40.34 - 25.45 hPa |
|----------|-------------------|
| Layer 9  | 25.45 - 16.06 hPa |
| Layer 10 | 16.06 - 0.13 hPa  |
| Layer 11 | 10.13 - 6.393 hPa |
| Layer 12 | 6.393 - 4.034 hPa |
| Layer 13 | 4.034 - 2.545 hPa |
| Layer 14 | 2.545 - 1.606 hPa |

Table 2. SBUV satellite ozone data coverage used in this study.

| Nimbus 7 SBUV  | 11/1978 - 05/1990 |
|----------------|-------------------|
| NOAA-9 SBUV/2  | 02/1985 - 01/1998 |
| NOAA-11 SBUV/2 | 01/1989 - 03/2001 |
| NOAA-14 SBUV/2 | 03/1995 - 09/2006 |
| NOAA-16 SBUV/2 | 10/2000 - 05/2014 |
| NOAA-17 SBUV/2 | 08/2002 - 03/2013 |
| NOAA-18 SBUV/2 | 07/2005 - 11/2012 |
| NOAA-19 SBUV/2 | 03/2009 - present |

 Lidar vs SBUV2(common with Lidar) at HP
 Lidar vs SBUV2(common with Lidar) at MLO
 Lidar vs SBUV2(common with Lidar) at OHP
 Lidar vs SBUV2(common with Lidar) at LAU
 Lidar vs SBUV2(common with Lidar) at TBL 59000 (a) 49500 (m) Height (m) Layer 14 (2.545 - 1.606 hPa) Layer 13 (4.034 - 2.545 hPa) Layer 12 (6.393 - 4.034 hPa) Layer 11 (10.13 - 6.393 hPa) 32000 Layer 10 (16.06 - 10.13 hPa) Layer 9 (25.45 - 16.06 hPa) 24000 Layer 8 (40.34 - 25.45 hPa) + 16500 1.0 0.0 0.2 0.4 0.6 0.8 Cor. Coef. (r) Lidar vs SBUV2(common with Lidar) at HP — Lidar vs SBUV2(common with Lidar) at MLO Lidar vs SBUV2(common with Lidar) at OHP - Lidar vs SBUV2(common with Lidar) at LAU Lidar vs SBUV2(common with Lidar) at TBL • 59000 49500 (m) 41000 Height Layer 14 (2.545 - 1.606 hPa) Layer 13 (4.034 - 2.545 hPa) Layer 12 (6.393 - 4.034 hPa) Layer 11 (10.13 - 6.393 hPa) 32000 Layer 10 (16.06 - 10.13 hPa) Layer 9 (25.45 - 16.06 hPa) 24000 Layer 8 (40.34 - 25.45 hPa) -+16500 1.0 0.0 0.2 0.4 0.6 0.8 Cor. Coef. (r)

Figure 1. (a) Correlation between monthly mean ozone anomalies from lidar and SBUV station overpasses on common days. Best correlations are between 25 and 32 km. All correlations are statistically significant at 99.99%. HP: Hohenpeißenberg, OHP: Haute Provence, TBL: Table Mountain, MLO: Mauna Loa, LAU: Lauder. (b) Same as in (a) but comparing monthly mean SBUV overpasses from about 30 days in a month with monthly mean SBUV overpasses from only days when lidar measurements were available. All correlations are statistically significant at 99.99%.

---

## Author Response (AR2)

Dear Mark,

We would like to thank you for your careful reading and corrections. All changes have been done as suggested.

p. 5, l. 164ff: "As to the role played by other sources of natural variability e.g. QBO, solar cycle etc., which were not removed from the time series in the correlations, it is small and does not exceed 2-4% of the explained variance as seen in a separate analysis (not shown here)." I do not understand this, the correlations are mainly determined by the collinearity of the variability in the compared timeseries. In other words, if there was no short-term variability (and after removing a fitted polynomial) correlations would be rather low. I suggest to remove this sentence.
**Answer: The sentence has been removed.**

p. 12, l. 439. Remove "Moving"
**Answer: Done**

Figure 3: Add as a vertical line the latitudes of the stations presented in each panel.
**Answer: Done**

[revised manuscript text omitted]